# Enzymatic kinetic resolution of desmethylphosphinothricin indicates that phosphinic group is a bioisostere of carboxyl group

Daniela De Biase [1✉], Francesca Cappadocio[1], Eugenia Pennacchietti[1], Fabio Giovannercole [1], Antonio Coluccia [2], Jouko Vepsäläinen [3] & Alex Khomutov [4✉]

*Escherichia coli* glutamate decarboxylase (*Ec*GadB), a pyridoxal 5′-phosphate (PLP)-dependent enzyme, is highly specific for *L*-glutamate and was demonstrated to be effectively immobilised for the production of γ-aminobutyric acid (GABA), its decarboxylation product. Herein we show that *Ec*GadB quantitatively decarboxylates the *L*-isomer of *D,L*-2-amino-4-(hydroxyphosphinyl)butyric acid (*D,L*-Glu-γ-P$_H$), a phosphinic analogue of glutamate containing *C-P-H* bonds. This yields 3-aminopropylphosphinic acid (GABA-P$_H$), a known GABA$_B$ receptor agonist and provides previously unknown *D*-Glu-γ-P$_H$, allowing us to demonstrate that *L*-Glu-γ-P$_H$, but not *D*-Glu-γ-P$_H$, is responsible for *D,L*-Glu-γ-P$_H$ antibacterial activity. Furthermore, using GABase, a preparation of GABA-transaminase and succinic semialdehyde dehydrogenase, we show that GABA-P$_H$ is converted to 3-(hydroxyphosphinyl)propionic acid (Succinate-P$_H$). Hence, PLP-dependent and NADP$^+$-dependent enzymes are herein shown to recognise and metabolise phosphinic compounds, leaving unaffected the *P-H* bond. We therefore suggest that the phosphinic group is a bioisostere of the carboxyl group and the metabolic transformations of phosphinic compounds may offer a ground for prodrug design.

[1] Department of Medico-Surgical Sciences and Biotechnologies, Laboratory Affiliated to Istituto Pasteur Italia—Fondazione Cenci Bolognetti, Sapienza University of Rome, Corso della Repubblica 79, I-04100 Latina, Italy. [2] Department of Chemistry and Technology of Drugs, Laboratory Affiliated to Istituto Pasteur Italia—Fondazione Cenci Bolognetti, Sapienza University of Rome, Piazzale Aldo Moro 5, I-00185 Rome, Italy. [3] School of Pharmacy, Biocenter Kuopio, University of Eastern Finland, Kuopio Campus, P.O. Box 1627, Kuopio FI-70211, Finland. [4] Engelhardt Institute of Molecular Biology, Russian Academy of Sciences, Vavilov St., 32, Moscow 119991, Russia. ✉email: daniela.debiase@uniroma1.it; alexkhom@list.ru

The pyridoxal 5′-phosphate (PLP)-dependent enzyme glutamate decarboxylase (L-glutamate 1-carboxy-lyase; Gad; E.C. 4.1.1.15) catalyses the irreversible decarboxylation on C-α of the proteinogenic amino acid L-glutamate to yield γ-aminobutyrate (GABA). The enzyme from *Escherichia coli* is active only on the L-isomer of glutamate, with the D-isomer neither being a substrate nor an inhibitor[1,2]. The *E. coli* enzyme, intensively studied at the biochemical and structural level, is a homohexamer, active in the pH range 3.8–5.4, positively responding to chloride ions, and undergoing a remarkable auto-inactivation above pH 5.5[3,4]. Lammens et al.[5] were the first to show that *E. coli* GadB (*Ec*GadB) can be efficiently immobilised and used as an excellent tool for the bio-based synthesis of GABA. This study was followed by several others[6–9]. The interest residing in the fact that L-glutamate, a non-essential amino acid and one of the most abundant in proteins, can be extracted from waste biomass and as an alternative source for the production of a valuable compound, such as GABA, from which 2-pyrrolidone, N-methyl-pyrrolidone, N-vinyl-pyrrolidone as well as polymers and pharmaceuticals (e.g., cotinine, piracetam, iodopovidone) can be derived[10,11]. In addition, *Ec*GadB was engineered to optimise GABA synthesis for large scale production in particular by rendering the enzyme less prone to rapidly auto-inactivate at pH > 5.5[4,11,12].

Even though *Ec*GadB decarboxylates at a significant rate only L-glutamate, its physiological substrate ($K_m \sim 1$ mM)[1,13], it was reported that the enzyme slowly decarboxylates glutamate analogues such as: L-glutamate-γ-methylester, *threo*-β-hydroxy-glutamate, 3-fluoro-L-glutamate, γ-methylene-glutamate, L-homocysteine sulphinic acid and L-homocysteic acid[1,14–17]. Except for the latter two, none of the listed compounds is a naturally occurring amino acid.

Naturally occurring amino acids also include some phosphonic and phosphinic compounds, i.e., containing a carbon–phosphorous bond (C–P). Phosphonates can mimic a phosphate monoester or a tetrahedral intermediate (or reaction transition states) of the carboxyl group transformations[18–20]. These unusual phosphorus-containing compounds, naturally produced by many micro-organisms, are synthesised chemically in the pharmaceutical and agro-chemical industry and have found application in medicine and agriculture[21,22]. Notable examples include the antibiotic fosfomycin, the antimalarial fosmidomycin and the herbicides bialaphos (PTT; Fig. 1) and phosalacine, which are tripeptides containing phosphinothricin (PT; Fig. 1) followed by alanyl-alanine and alanyl-leucine residues, respectively. PT is a glutamate analogue known to inhibit glutamine synthetase by

mimicking the phosphorylated intermediate of glutamate formed during the enzymatic reaction[23,24]. Bialaphos was first shown to be produced by *Streptomyces viridochromogenes* and *Streptomyces hygroscopicus*[25,26]. In both micro-organism the biosynthetic pathway leading to bialaphos was later characterised[27]. Along this pathway, L-2-amino-4-(hydroxyphosphinyl)butyric acid (also known as desmethylphosphinothricin, DMPT; Fig. 1), hereafter L-Glu-γ-P_H, was shown to be a key intermediate, becoming detectable only in some mutant strains[28].

L-Glu-γ-P_H is a phosphinic analogue of glutamate and to date very little is known about its biological activity: when tested for its in vitro ability to bind group III metabotropic L-Glu receptors, which are mostly presynaptic and controlling glutamate release at the synaptic level, L-Glu-γ-P_H potency was found in the same range as that of glutamate[29]. Recent work carried out in our laboratories suggests that L-Glu-γ-P_H has antibacterial activity[30]. Except for these reports, the substrate/inhibitor properties of L-Glu-γ-P_H on glutamate metabolising enzymes, including bacterial glutamate decarboxylase, have never been studied.

In the present work, we investigate the substrate properties of L-Glu-γ-P_H on *Ec*GadB and, by exploiting the specificity of the enzyme for the L-isomer, develop a bio-based route for the kinetic resolution of the D-isomer from D,L-2-amino-4-(hydroxyphosphinyl)butyric acid (D,L-Glu-γ-P_H). The efficient enzymatic decarboxylation of L-Glu-γ-P_H yields 3-aminopropylphosphinic acid (Fig. 1), hereafter GABA-P_H, an agonist of GABA_B receptor shown to be more potent than Baclofen in vitro[31]. Moreover, we demonstrate that GABA-P_H can be further metabolised to yield the phosphinic analogue of succinate (Succinate-P_H; Fig. 1).

## Results

**Retrosynthetic analysis of D-Glu-γ-P_H.** The earlier unknown D-Glu-γ-P_H (**1**) may be prepared chemically (Fig. 2) and key precursor is methyl N-Cbz-D-vinylglycine (**2**), which can be synthesised following the protocols developed and used for the corresponding L-isomer[32]. To prepare methyl N-Cbz-L-vinylglycine different starting compounds, including amino acids, were used and reported elsewhere[32] and many of these synthetic approaches may be directly applied to prepare (**2**). In brief, the most well-known pathways are shown as B, C, D in Fig. 2. Pathway (D), starting from D-Met, follows a classical and widely used 3-step Rapoport synthesis[32]. Alternatively, (**2**) can be prepared from commercially available D-3-aminobutyrolactone hydrochloride (**3**) in six steps, either following pathway (C) via selenide (**4**)[33] or via sulfoxide (**5**), following pathway (B), which also consists of six steps[34]. Besides, pathway (C) requires working with diphenyldiselenide, which is toxic for vertebrates and environment[33].

Conversion of (**2**) into target D-Glu-γ-P_H (**1**) consists in anti-Markovnikov radical addition of $H_3PO_2$ to the double bond of (**2**) and subsequent removal of the protecting groups, as described for L-Glu-γ-P_H in Supplementary Methods and elsewhere[29]. Hence, chemical syntheses of D-Glu-γ-P_H, are laborious, consist of five to eight steps and can provide target D-Glu-γ-P_H at a rather low overall yield.

The stereoselectivity of *Ec*GadB could alternatively be exploited for the kinetic resolution of D,L-Glu-γ-P_H using *Ec*GadB (Fig. 2, pathway A) as a fast, quantitative and inexpensive bio-based approach to synthesise unknown D-Glu-γ-P_H and, at the same time, to obtain GABA-P_H, the anticipated decarboxylation product of L-Glu-γ-P_H by *Ec*GadB. This working hypothesis was based on the assumption that the single-charged phosphinic group mimics the flat carboxyl group not only in the global charge (i.e., single), but also because of its flattened tetrahedral conformation (Supplementary Fig. 1) as suggested by the

**Fig. 1 Chemical structures of DMPT and its metabolically related derivatives.** The herbicide bialaphos (PTT), produced by *S. viridochromogenes* and *S. hygroscopicus*, is a tripeptide containing L-phosphinothricin (PT) followed by alanyl-alanine (L-Ala-LAla) residues. In the biosynthetic pathway leading to PTT, desmethylphosphinothricin (DMPT; L-Glu-γ-P_H) is a key intermediate, from which 3-aminopropylphosphinic acid (3-APPA; GABA-P_H) and Succinate-P_H can be derived following in order DMPT decarboxylation and oxidation, respectively.

**Fig. 2 Retrosynthetic analysis of the D-Glu-γ-P_H.** (A) From racemic D,L-Glu-γ-P_H (this work, in blue) in one enzymatic step; (B) from D-2-aminobutyrolactone hydrochloride (**3**) via o-nitrophenyl-sulfoxide (**5**) in six steps; (C) from D-2-aminobutyrolactone hydrochloride (**3**) via phenylselenide (**4**) in six steps; (D) from D-methionine via sulfoxide (**6**) in three steps. Pathways (B)–(D) lead to the same key precursor methyl N-Cbz-D-vinylglycine (**2**).

crystallographic evidence on the bond lengths in the β-phosphinic analogue of aspartate[35].

Given that D,L-Glu-γ-P_H is readily available with high yield via four steps[36] or via a three-step one-pot synthesis[37] from dibutyl ester of vinylphosphonic acid, the bioconversion protocol, herein used, is proposed to leave D-Glu-γ-P_H unmodified from D,L-Glu-γ-P_H, assuming that the L-isomer is quantitatively converted into GABA-P_H by EcGadB. The downstream purification from the reaction mixture would therefore allow to prepare unknown D-Glu-γ-P_H and commercially unavailable, biologically active GABA-P_H.

**Preliminary studies on L-Glu-γ-P_H as substrate of EcGadB.** Substrate properties of L-Glu-γ-P_H (DMPT; Fig. 1) in EcGadB reaction were not self-evident, since phosphinothricin (PT; Fig. 1) and the γ-phosphonic counterpart of L-Glu-γ-P_H, i.e. 2-amino-4-phosphonobutyric acid, were shown to be neither substrates nor inhibitors of the enzyme[38,39]. Initially, we investigated if EcGadB can bind D,L-Glu-γ-P_H and specifically decarboxylate it into GABA-P_H by assessing the course of the reaction using thin layer chromatography (TLC). These experiments demonstrated the formation of a compound having the $R_f$ value similar to that of GABA (Supplementary Fig. 2). To confirm that the newly formed compound is GABA-P_H, we performed [1]H NMR analysis of EcGadB reaction mixture with D,L-Glu-γ-P_H as a substrate in Pyridine/HCl buffer. Glu-γ-P_H and GABA-P_H have characteristic signals that only partly overlap in [1]H NMR spectra (Supplementary Fig. 3) that clearly showed that the incubation of D,L-Glu-γ-P_H with EcGadB results in the formation of GABA-P_H (Supplementary Fig. 3). In addition, TLC provided additional evidence that the γ-phosphonic counterpart of L-Glu-γ-P_H is not a substrate of EcGadB, as already reported[38,39].

**Modelling of L-Glu-γ-P_H binding in EcGadB active site.** To compare the binding mode and interactions of L-Glu and L-Glu-γ-P_H to EcGadB active site, molecular dynamics (MD) simulations[40] were carried out by setting a constant pH of 4.5 and using the Protein Data Bank code 1PMM[41]. The external aldimine adducts PLP-L-Glu and PLP-L-Glu-γ-P_H were drawn using the coordinates of the co-crystallised PLP and acetic acid (ACY) in 1PMM[41]. The trajectory analyses for the L-Glu with its cognate protein showed a good stability with a root mean square fluctuation (RMSF) of 1.12, 0.46 and 0.76 Å for the enzyme, the PLP-Lys276 internal aldimine and the PLP-L-Glu external aldimine adduct, respectively. The inspection of the binding modes highlighted a series of key contacts that are characteristic of the PLP-L-Glu adduct into the binding site: the PLP moiety, which was superimposable with the binding conformation reported in 1PMM, is stabilised by hydrophobic contact with Gln163 and

Ala245 side chains. Furthermore, the PLP hydroxyl group is engaged in intramolecular H-bond with the aldimine nitrogen atom. The phosphate group is involved in H-bonds network: the OP1 with Ser127 and Ser128, the OP2 with Ser273* and His275, the OP3 with Ser318*. The α-carboxyl group is perpendicular to the PLP plane and too far (4.6 Å) from Gln163 to establish a polar contact. The γ-carboxylic moiety is engaged in a network of H-bonds involving Asp86*, Phe63 and Cys64, in good accordance with the literature data (Fig. 3, left panel)[41].

The same analyses were performed for L-Glu-γ-P_H. The RMSF values were 1.35, 0.47 and 1.07 Å for the enzyme, the PLP-Lys276 internal aldimine and the PLP-L-Glu-γ-P_H external aldimine, respectively. The contacts that stabilised the PLP-L-Glu-γ-P_H adduct (Fig. 3, right panel) resembled the interactions already reported for the PLP-L-Glu (Fig. 3, left panel). Also, the polar contacts that stabilised the phosphinic moiety of L-Glu-γ-P_H were similar, though not identical, to those described for the L-Glu γ-carboxylic acid and involving Thr62, Phe63, Cys64 and Asp86*. It was in addition noticed that during the simulations the distances between the Lys276 ε-amino group and the external aldimine C-α atom of the amino acid substrate were always close to 3.5 Å, in agreement with the catalytic role of this residue[42]. Together with the preliminary evidence (Supplementary Figs. 2 and 3), the good superimposition between the L-Glu and L-Glu-γ-P_H binding modes (Fig. 3) strengthens the working hypothesis that the L-Glu-γ-P_H is an EcGadB substrate. However, some noticed differences may account for the observed changes in the binding affinity (next section). Noteworthy, the same sort of analysis failed to provide a reliable model for the binding of the γ-phosphonic counterpart of L-Glu-γ-P_H to the EcGadB active site. Thus supporting previous studies[36,37] and the TLC analysis (Supplementary Fig. 2).

**L-Glu-γ-P_H is a substrate of EcGadB and GABA-P_H of GABase.** L-Glu-γ-P_H is not commercially available. In this work it was synthesised according to a published protocol[29], with some modifications (Supplementary Methods), and, supported by the preliminary data described in the previous section, its substrate properties on EcGadB were studied.

In order to obtain the kinetic parameters of the EcGadB reaction with L-Glu-γ-P_H, we initially investigated whether the GABase assay, routinely used to measure GABA formed following the decarboxylation of L-Glu by bacterial EcGadB activity[13,42,43], could be employed also for detecting the products of the reaction with L-Glu-γ-P_H. GABase is a commercial crude preparation of PLP-dependent GABA-transaminase and NADP[+]-dependent succinic semialdehyde dehydrogenase from Pseudomonas fluorescens: it works in the neutral-to-mild alkaline pH range by converting GABA into succinic semialdehyde

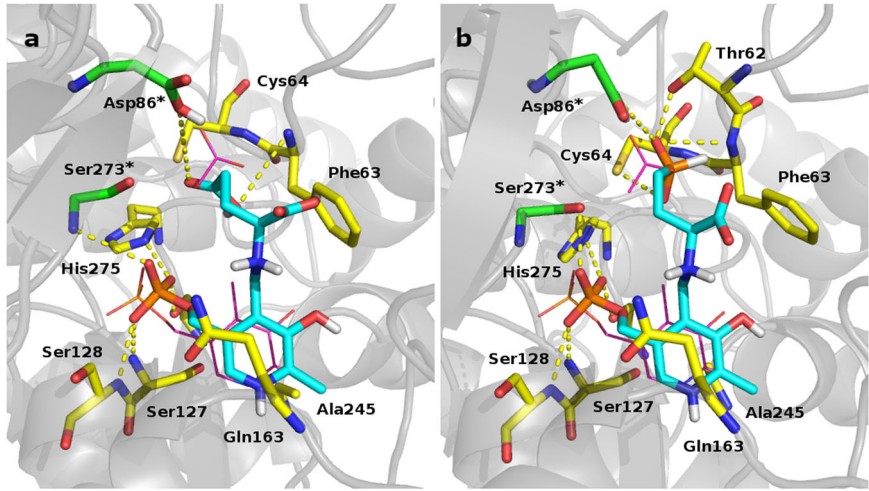

**Fig. 3 Proposed binding mode for studied adducts.** Snapshots of the molecular dynamic simulations. **a** Proposed binding mode for PLP-*L*-Glu adduct (cyan). **b** Proposed binding mode for PLP-*L*-Glu-γ-P$_H$ adduct (cyan). Residues involved in interaction are reported as stick, yellow for chain B and green for chain A of the functional dimer[41]. PLP and acetic acid (ACY) of 1PMM are also reported as magenta lines. The enzyme is depicted as grey cartoon and the H-bonds are reported as yellow dotted lines. Active site residues contributed by the other subunit in the functional dimer are also labelled with an asterisk.

(via GABA-transaminase activity)[43,44] which is then irreversibly oxidised under the assay conditions into succinic acid (via succinic semialdehyde dehydrogenase) with quantitative formation of NADPH. The latter reduced cofactor can be easily detected spectrophotometrically[13,45,46]. It was therefore of interest to find out whether GABA-P$_H$, generated by the irreversible decarboxylation of *L*-Glu-γ-P$_H$, could be detected through GABase. Succinate semialdehyde-P$_H$ (SSA-P$_H$), the phosphinic analogue of succinic semialdehyde and transamination product of GABA-P$_H$, has in its structure a phosphinic moiety with a P–H bond and a carbonyl group, both possessing a high reducing potential. In order to assess to which one of the two groups (phosphinic or carbonyl) the formation of NADPH could be assigned, $^{31}$P NMR spectroscopy analysis was performed before and after GABase treatment of the *Ec*GadB reaction mixture (Fig. 4) as follows: (1) the conversion of *D,L*-Glu-γ-P$_H$ (spectrum **a** in Fig. 4) into GABA-P$_H$ (spectrum **b** in Fig. 4) was allowed to occur in the presence of *Ec*GadB, at pH 4.6, and spectrum **b** was run using half of the reaction mixture, after removing *Ec*GadB by ultrafiltration; (2) the remainder of the reaction mixture was dried under vacuum and resuspended in an isovolume of GABase assays solution to allow GABA-P$_H$ to be converted into Succinate-P$_H$ by GABase[46].

For comparative purposes, the latter reaction was terminated before completion to still allow detection of the GABA-P$_H$ signal in the final $^{31}$P NMR spectrum. Before running the latter $^{31}$P NMR spectroscopy analysis, GABase was removed by ultrafiltration. The signals of the three compounds present after the GABase reaction are in the region typical for the expected phosphinic compounds (spectrum **c** in Fig. 4) and the identity of each was further corroborated by $^{31}$P NMR spectroscopy analysis of a mixture of model compounds (spectrum **d** in Fig. 4).

Thus, the $^{31}$P NMR data provided incontrovertible evidence that the formation of NADPH, as detected spectrophotometrically (data not shown), was due to the oxidation of the carbonyl moiety of 3-(hydroxyphosphinyl)propanal (SSA-P$_H$ in Fig. 4) yielding Succinate-P$_H$. To the best of our knowledge, while previously hypothesised[47], this is the first report where direct evidence of a selective NAD(P)$^+$-dependent enzymatic oxidation of a functional group in the side chain of a phosphinic acid, which leaves unaffected the P–H bond, is provided.

The substrate properties of *L*-Glu-γ-P$_H$ for *Ec*GadB were therefore assessed by kinetic analysis, using GABase which allowed to quantitatively measure the GABA-P$_H$ produced by detecting spectrophotometrically the NADPH produced. These results were compared with those obtained with *L*-Glu, the physiological substrate of *Ec*GadB (Table 1).

Based on the results shown in Table 1, it turned out that *Ec*GadB is >200 folds less efficient in decarboxylating *L*-Glu-γ-P$_H$ if compared with *L*-Glu. This difference could be primarily assigned to the increase in $K_m$ (Table 1) and explained, at least in part, by some differences in the anchoring of the distal carboxyl and phosphinic groups in *Ec*GadB active centre, as suggested by the modelling studies (Fig. 3).

**Enantioselective decarboxylation of *D,L*-Glu-γ-P$_H$ by *Ec*GadB.**
In the previous sections, evidence has been provided that *Ec*GadB can accept *L*-Glu-γ-P$_H$ as a substrate and decarboxylate it into GABA-P$_H$. At the same time, pathway A in Fig. 2 shows that *D*-Glu-γ-P$_H$ could be easily prepared starting from *D,L*-Glu-γ-P$_H$, by taking advantage of the well-known selectivity of the enzyme for the *L*-isomer of glutamate[1,2], which would therefore yield GABA-P$_H$ and leave unchanged *D*-Glu-γ-P$_H$.

In order to assess the feasibility of this approach, a preliminary experiment (on a small scale; 100 μL) using *Ec*GadB and *D,L*-Glu-γ-P$_H$ (0.4 mol L$^{-1}$) as substrate was carried out for 2 h, under the conditions described in the "Methods" section. The production of GABA-P$_H$ followed hyperbolic kinetics and the amount present at 2 h (0.197 mol L$^{-1}$) was almost quantitative, according to the GABase assay, with respect to the amount (0.200 mol L$^{-1}$) of *L*-isomer present in the original reaction mixture. The kinetics of GABA-P$_H$ formation using GABase (Supplementary Fig. 4) indicated that close to 50% of *D,L*-Glu-γ-P$_H$ remained unchanged. This provided a strong indication that indeed *Ec*GadB could decarboxylate only the *L*-isomer of Glu-γ-P$_H$ and therefore be exploited to obtain preparative amounts of *D*-Glu-γ-P$_H$ and GABA-P$_H$. The key parameter of this enzymatic reaction, as already reported for the preparative synthesis of GABA from *L*-Glu[5,12], was pH, which was controlled at time intervals in order to be within the range 4.6–5.0, where *Ec*GadB displays maximal activity[12,13].

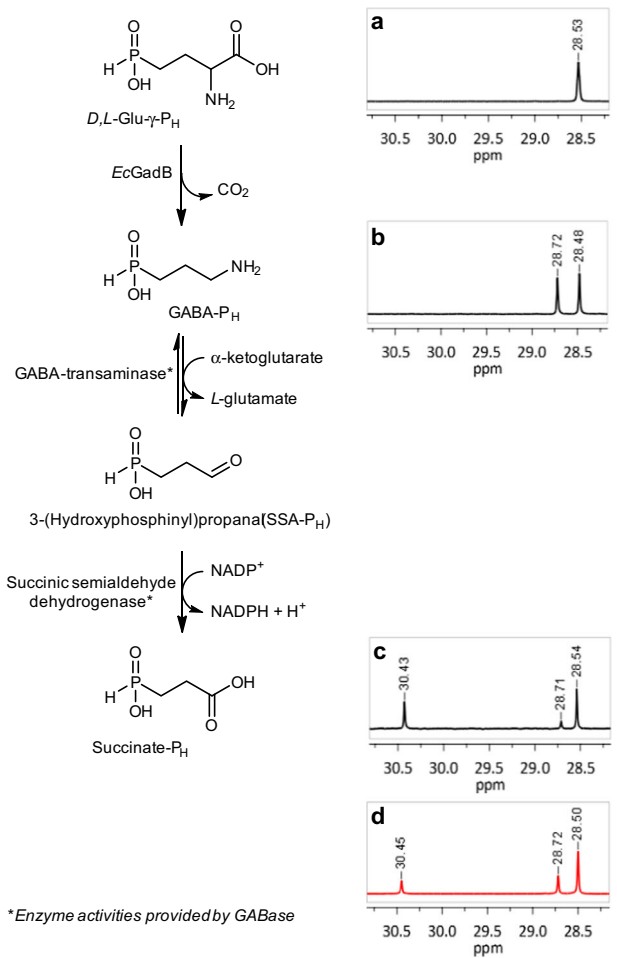

**Fig. 4 Enzymatic reactions that yield Succinate-$P_H$ and NADPH starting from $D,L$-Glu-$\gamma$- $P_H$.** The $Ec$GadB (0.09 mg ml$^{-1}$) decarboxylation reaction of $D,L$-Glu-$\gamma$- $P_H$ (18 mM) was carried out at pH 4.6, in 10 mM sodium acetate buffer (pH adjusted with small additions of diluted HCl) and then halted after 4 h by removing $Ec$GadB by ultrafiltration. Half of the flow-through was dried and resuspended in an isovolume of GABase assay solution where the two indicated enzymatic reactions were run one-pot, with succinic semialdehyde dehydrogenase activity making the whole process irreversible. $^{31}$P NMR spectra of: (**a**) $Ec$GadB reaction mixture at time zero; (**b**) $Ec$GadB reaction mixture after interruption by enzyme removal; (**c**) reaction mixture after GABase treatment; (**d**) model mixture of $L$-Glu-$\gamma$-$P_H$, GABA-$P_H$, and Succinate-$P_H$.

**Table 1 Kinetic parameters of $L$-Glu and $L$-Glu-$\gamma$-$P_H$ using $Ec$GadB[a].**

| Substrate | $k_{cat}$ (s$^{-1}$) | $K_m$ (mM) | $k_{cat}/K_m$ (s$^{-1}$ mM$^{-1}$) |
|---|---|---|---|
| $L$-Glu | 78.18 ± 2.06 | 0.84 ± 0.09 | 92.85 ± 10.26 |
| $L$-Glu-$\gamma$-$P_H$ | 13.93 ± 0.58 | 39.82 ± 5.66 | 0.35 ± 0.05 |

[a]Data ± standard deviation were obtained by fitting the experimental points through an hyperbolic curve using the Michaelis–Menten equation.

Preparative decarboxylation was therefore set up by scaling up the above reaction 25 times (final volume 2.5 mL). In this experiment $^{31}$P NMR spectroscopy was employed in a real-time mode to allow a direct and immediate detection of GABA-$P_H$ formation from $D,L$-Glu-$\gamma$-$P_H$. As expected, the amount of Glu-$\gamma$-$P_H$ in the reaction mixture decreased with time and a

concomitant increase of GABA-$P_H$ was observed, which after 3 h reached 93% of the theoretical value (Fig. 5).

The reaction was allowed to proceed for additional 2 h during which the residual $L$-Glu-$\gamma$-$P_H$ was decarboxylated and the amount of GABA-$P_H$ reached 98.2% of the theoretical value.

The preparative experiment was performed in triplicate, with kinetics and final yields closely resembling those shown in Fig. 5.

These results provided a strong indication that indeed half of $D,L$-Glu-$\gamma$-$P_H$ was not decarboxylated (very likely the $D$-isomer) and that the bioconversion could be employed to obtain preparative amounts of GABA-$P_H$ and $D$-Glu-$\gamma$-$P_H$, as confirmed by physical-chemical techniques and enzymatic assays (next section).

**Isolation of enzymatically obtained $D$-Glu-$\gamma$-$P_H$ and GABA-$P_H$.** After the removal of $Ec$GadB through ultrafiltration, GABA-$P_H$ and $D$-Glu-$\gamma$-$P_H$ (according to the enantioselectivity of the enzyme) were isolated from the reaction mixture using ion-exchange chromatography on Dowex 50WX8 (H$^+$-form). Elution with water provided crude $D$-Glu-$\gamma$-$P_H$ which was contaminated with PLP, originally present in the reaction mixture (see "Methods" section). GABA-$P_H$ hydrochloride was recovered with 0.5 N HCl. It was then converted to zwitterion and recrystallised. This sample was identical to chemically synthesised GABA-$P_H$ as assessed by $^1$H, $^{31}$P and $^{13}$C NMR spectra (Supplementary Fig. 5), TLC and melting point (see "Methods" section).

In order to separate $D$-Glu-$\gamma$-$P_H$ and PLP, the above crude sample was treated with an excess of aminooxyethyl putrescine (AOEPUT) as depicted in Supplementary Fig. 6. AOEPUT does not react with Glu-$\gamma$-$P_H$, but forms oxime with PLP fast and quantitatively. The two amino groups of both free AOEPUT and AOEPUT-PLP oxime ensure tight binding to Dowex 50WX8, making it impossible to elute them with water, but on the contrary allowed to recover $D$-Glu-$\gamma$-$P_H$ from the resin. After crystallisation, $D$-Glu-$\gamma$-$P_H$ was analysed by $^1$H, $^{31}$P and $^{13}$C NMR spectroscopy (Supplementary Fig. 7) and displayed $[\alpha]_D^{20}$ value of $-20.4°$, ($c$ 1.0, H$_2$O). The direction of the rotation clearly indicated it to be the $D$-isomer (see "Methods" section) because its value was in full agreement with the rotation $[\alpha]_D^{20}$ value of $+20.9°$, ($c$ 1.0, H$_2$O) of $L$-Glu-$\gamma$-$P_H$ chemically synthesised in this work (Supplementary Methods). Notably, the latter value is rather different from that reported in the literature for $L$-Glu-$\gamma$-$P_H$ $[\alpha]_D^{20}$ $+7.9°$ ($c$ 0.4, H$_2$O)[29]. Thus the herein enzymatically prepared $D$-Glu-$\gamma$-$P_H$ has the same specific rotation value of the chemically synthesised $L$-Glu-$\gamma$-$P_H$, though in the opposite direction. This latter observation provides an unquestionable indication that the two isomers prepared in this work are indeed the exact compounds. To further prove this, the reactivity of $D$-Glu-$\gamma$-$P_H$ towards $Ec$GadB was studied. Any attempt to detect GABA-$P_H$ from $D$-Glu-$\gamma$-$P_H$ failed, even when the reaction was allowed to occur overnight, in line with the well-known stereospecificity of the enzyme (data not shown).

**Antimicrobial activity of Glu-$\gamma$-$P_H$ enantiomers.** To assess the contribution of each isomer to the antimicrobial activity of $D,L$-Glu-$\gamma$-$P_H$[30], the minimum inhibitory concentration (MIC) of the racemate as well as that of the $L$- and $D$-isomers were assayed on the $E. coli$ K12 strain MG1655 (Table 2).

The results in Table 2 provided preliminary, but clear indication that the $L$-isomer was the only molecule responsible for the antibacterial activity and that its MIC is in the same range of a well-known antibiotic, i.e., ampicillin. Notably, to reach MIC$_{90}$, the concentration of $D,L$-Glu-$\gamma$-$P_H$, had to be twice as high as that of $L$-Glu-$\gamma$-$P_H$, thus ensuring that the $L$-isomer was at the effective working concentration. The MIC value of the

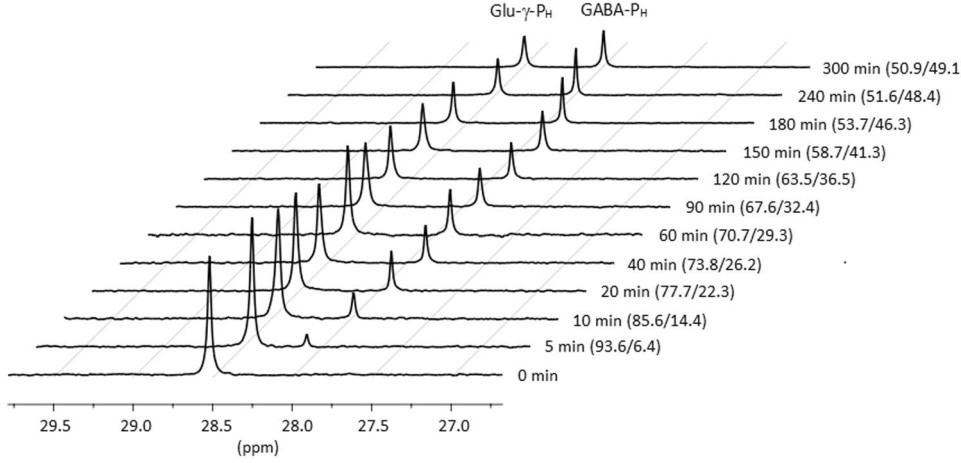

**Fig. 5 $^{31}$P NMR spectroscopy analysis.** The kinetics of the decarboxylation of $D,L$-Glu-γ-P$_H$ by $Ec$GadB yielding GABA-P$_H$ are shown. The time point (in minutes) at which each measurement was done is shown on the right and next to it (in brackets) is provided the Glu-γ-P$_H$/GABA-P$_H$ ratio obtained by peak integration. Fragments of the $^{31}$P NMR spectra are shown as a stack.

**Table 2 Minimum inhibitory concentration on *E. coli* K12.**

| Compound | MIC$_{90}$ (µg mL$^{-1}$) |
|---|---|
| $D,L$-Glu-γ-P$_H$ | 14–20 |
| $L$-Glu-γ-P$_H$ | 7 |
| $D$-Glu-γ-P$_H$ | nd[a] |
| Ampicillin | 4 |

[a]*nd* not detectable.

racemate is thus compatible with one isomer ($L$) being active and the other one ($D$) being inactive. The interval in the MIC$_{90}$ of the racemate suggests that the $D$-isomer may slightly interfere with the $L$-isomer transport, a finding non uncommon in transport studies[48]. On the other hand, the $D$-isomer, when assayed alone, exhibited a hardly detectable activity, i.e., it caused only 4% of inhibition of growth at the concentration of 128 µg mL$^{-1}$. Thus, the MIC analysis confirmed that via kinetic resolution the distomer, i.e. $D$-Glu-γ-P$_H$, could be obtained through a bioconversion without the need of chemically synthesising it and it was devoid of a significant antibacterial activity.

## Discussion

Through a combination of enzymology, NMR spectroscopy, molecular modelling and microbiological assays, we provide evidence that in some phosphinic compounds, i.e. containing C–P–H bonds, the phosphinic moiety acts as a bioisostere of the carboxyl group, making them substrates of the relevant enzymes. In particular, as part of the kinetic resolution of $D,L$-Glu-γ-P$_H$ using $Ec$GadB, it is possible to prepare unknown $D$-Glu-γ-P$_H$ and biologically active GABA-P$_H$ and offer an excellent route to their synthesis.

Our working hypothesis was based on the assumption that the single-charged phosphinic group mimics the flat carboxyl group not only in the global charge (i.e., single) but also because of its flattened tetrahedral conformation (Supplementary Fig. 1). On the contrary, the double-charged tetrahedral phosphonic group turns out to be a bad mimetic of the carboxyl group (Supplementary Fig. 1). Our hypothesis was also in good agreement with previous reports showing that (1) the γ-phosphonic analogue of $L$-Glu, as well as PT, is neither substrates, nor inhibitor of GadB[38,39]; (2) α-phosphinic analogues of amino acids, but not α-phosphonic analogues, are substrates of PLP-dependent alanine transaminase, methionine γ-lyase and tyrosine

phenol-lyase[47,49,50]; (3) that group III metabotropic $L$-Glu receptors display an affinity towards $L$-Glu-γ-P$_H$ comparable to that of $L$-Glu[29] and (4) $D,L$-Glu-γ-P$_H$ can act as a weak amino donor to the α-keto precursor of PT in *E. coli* GABA-transaminase[51].

Starting $D,L$-Glu-γ-P$_H$ is readily available with high yield[36,37], and the downstream purification from the bioconversion reaction mixture provides $D$-Glu-γ-P$_H$ with a preparative yield of 73%. Besides, as part of the kinetic resolution, GABA-P$_H$, an agonist of GABA$_B$ receptor shown to be more potent than Baclofen in vitro[31], is formed. In analogy with previously published work on the bioconversion of $L$-Glu with immobilised $Ec$GadB[5], preliminary data in solution indicate that the bioconversion reaction of $D,L$-Glu-γ-P$_H$ is feasible also in water (D. De Biase, personal communication), by controlling the pH of the reaction with a pH-stat.

Taking into account that $L$-Glu-γ-P$_H$ and some of its derivatives are currently under investigation as antibacterials[30], it was of key importance to assess that the $D$-Glu-γ-P$_H$, the distomer, was neither active nor displayed opposite or significantly negative effects in the in vivo testing. Moreover, the scalable bioconversion protocol described in this work makes $D$-Glu-γ-P$_H$ easily accessible to study its properties in relation to other currently unforeseen applications.

It is also important to recall that the substrate properties of the phosphinic analogue of succinic semialdehyde (SSA-P$_H$) as occurring in the GABase reaction were not self-evident, since it is known that phosphorous acid ($H_3PO_3$), having P–H bond, is efficiently oxidised to phosphoric acid ($H_3PO_4$) by NAD$^+$-dependent dehydrogenase encoded by the $ptxD$ gene[52]. Therefore, our results with *P. fluorescens* GABase confirm that in addition to the PLP-dependent enzyme (GABA-transaminase), a NADP$^+$-dependent dehydrogenase (Succinic semialdehyde dehydrogenase) can accept a phosphinic compound as substrate. The latter represents the first example of the NADP$^+$-dependent enzymatic transformation in the side chain of alkylphosphinic acids having the P–H bond with high reducing potential.

Our findings, summarised in Fig. 6, suggest that enzymatic approaches to the synthesis of phosphinic compounds are feasible.

Our data suggest that phosphinic compounds can be metabolised intracellularly to such a level that yields metabolites which eventually target enzymes different from the initially processed aminophosphinate; this may turn out to be a productive strategy

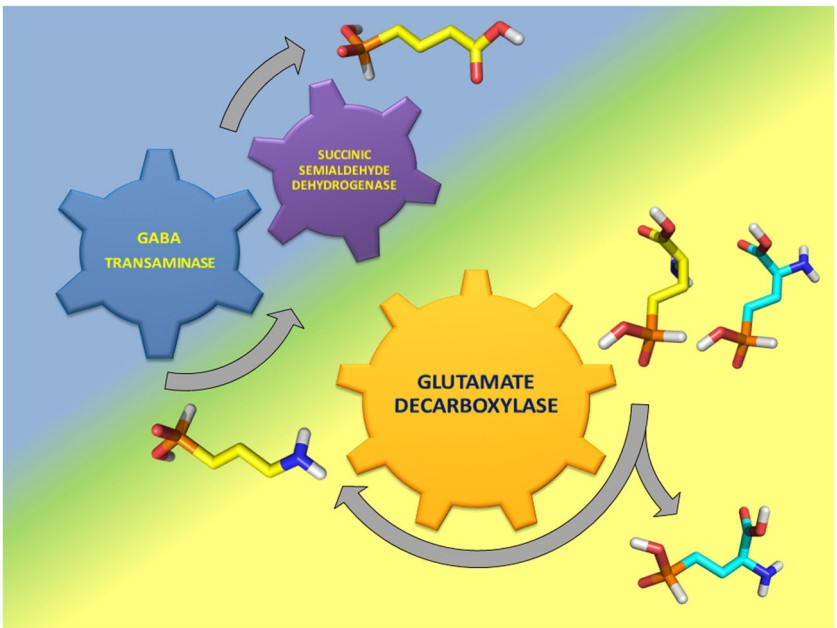

**Fig. 6 Phosphinic compounds as substrates of bacterial enzymes.** *Ec*GadB converts at mildly acidic pH (yellow background) the *L*-isomer of *D,L*-Glu-γ-P$_H$ (carbon skeleton in yellow on the outmost right), quantitatively yielding GABA-P$_H$ (carbon skeleton in yellow on the bottom left) and leaving *D*-Glu-γ-P$_H$ unmodified (carbon skeleton in cyan on the outmost right). The former becomes a substrate of GABase that yield 3-(hydroxyphosphinyl)propionic acid (i.e., Succinate-P$_H$ shown with carbon skeleton in yellow on the top centre) in a coupled reaction that occurs at mildly alkaline pH (blue background).

in prodrug design. To date, the only known example of this process is the intracellular transformation of the α-phosphinic derivative of *L*-Ala (Ala-P$_H$) into the phosphinic analogue of pyruvate, a well-characterised nM inhibitor of pyruvate dehydrogenase[53], with antitumor potential[20]. Thus, respect to compounds containing bonds C–P–C (i.e., methylphosphinic) or C–P–O (phosphonic), thoroughly investigated as such and as a key element of peptidomimetics[19], the biochemical and medicinal potential of compounds having C–P–H (phosphinic) bonds has received very little attention.

## Methods

**Materials**. *D,L*-Glu-γ-P$_H$ was synthesised as described earlier[37]. Synthesis of *L*-Glu-γ-P$_H$ was performed following a published protocol[29], with some modifications, as described in Supplementary Information. 3-(Hydroxyphosphinyl) propionic acid (Succinate-P$_H$) was synthesised as described elsewhere[54] and the product was purified from contaminating H$_3$PO$_2$ by column chromatography on Kieselgel 60 (40–63 μm, Merck, Germany), using CHCl$_3$:MeOH:AcOH, 7:2:1, v/v for elution. *N*-(2-aminooxyethyl)-putrescine trihydrochloride (AOEPUT) was synthesised as previously described[55]. Ion-exchange chromatography was carried out on Dowex 50WX8 (H$^+$-form, 100–200 mesh; BioRad, USA). Elution conditions are specified in the text. TLC was carried out on Kieselgel 60 F$_{254}$, or Cellulose F$_{254}$ plates (Merck, Germany) in *i*-PrOH:25%NH$_4$OH:H$_2$O, 7:1:2, v/v. Amino acids were detected on TLC plates following ninhydrin staining (0.4% in acetone).

For the expression, purification and assay of *Ec*GadB, ingredients and chemicals were from the following suppliers: the ingredients for bacterial growth were from BD-Difco (Becton, Dickinson and Company); streptomycin sulfate was from US Biochemical Corp. (Cleveland, OH, USA); ampicillin was from Roche; ion-exchange chromatography was performed using pre-packed HiPrep DEAE FF 16/10 (20 mL) from GE Healthcare Life Sciences; pyridine was from Carlo Erba; Vitamin B6, potassium dihydrogen phosphate, dipotassium hydrogen phosphate and kanamycin were from Fluka; sodium glutamate, sodium acetate, acetic acid, sodium chloride, GABA and PLP were purchased either from Merck or from Sigma-Aldrich. GABase was from Sigma-Aldrich. Vivaspin (4 mL and 500 μL; 30 and 10 kDa cutoff, respectively) ultrafiltration devices were from Sartorius Stedim Lab Ltd, UK.

For antimicrobial testing, Minimal Medium E[56], supplemented with 0.4% glucose, was used. Glucose was from Carlo Erba. All other ingredients were either from VWR or Applichem. The *E. coli* strain tested was the K12 strain MG1655[57,58].

**Analytical techniques**. $^1$H, $^{13}$C and $^{31}$P NMR spectra of *D*-Glu-γ-P$_H$ and GABA-P$_H$ were recorded on a Bruker 300 AM-300, while for NMR analysis of enzymatic reactions a Bruker 500 UltraShield$^{TM}$ was used. Spectra were measured in D$_2$O, or in

H$_2$O/D$_2$O mixture, by using with sodium 3-trimethyl-1-propanesulfonate as internal standard, or 85 % H$_3$PO$_4$ as an external standard. Chemical shifts are given in ppm. Optical rotations were recorded on a 341 Polarimeter (Perkin-Elmer); solvents and concentrations are indicated in the text. Melting points were determined in open capillary tubes on Electrothermals Mel-Temp 1202D instrument and are uncorrected.

**Enzymatic assays and calculation of kinetic parameters**. To find out the reaction conditions for the preparative scale experiments, the $k_{cat}$ and $K_m$ values for *L*-Glu were obtained by following *Ec*GadB (0.55 μg) reaction in the presence of *L*-Glu (from 0.156 to 20 mM) in a final reaction volume of 250 μL of 0.1 M Pyridine/HCl, pH 4.6 containing 0.1 mM DTT and 40 μM PLP. The experiment was performed in duplicate. The $k_{cat}$ and $K_m$ values for *L*-Glu-γ-P$_H$ were then determined in the same buffer system. In the latter case, GadB (13 μg) was incubated with *L*-Glu-γ-P$_H$ (from 6 to 220 mM) in a final reaction volume of 160 μL. The experiment was performed in duplicate. For both substrates, the rates were determined by halting the reaction, which consisted in transferring aliquots (50 or 25 μL) of the reaction mixture into four volumes (200 or 100 μL, respectively) of 0.1 M HEPPS buffer, pH 8.6, immediately followed by vigorous vortexing. At each tested concentration, three time points (for *L*-Glu 1–2–3 min; for *L*-Glu-γ-P$_H$ 3–7–10 min) were taken.

GABA (from *L*-Glu) or GABA-P$_H$ (from *L*-Glu-γ-P$_H$) content was assayed with the GABase assay by transferring 10 or 50 μL of each halted reaction into 100 μL of GABase solution containing 1.25 μL of activated GABase (0.02 Units μL$^{-1}$). The incubation was carried out for 60 min at 37 °C to allow the full transamination and oxidation by GABase of GABA (or GABA-P$_H$) into succinate (or Succinate-P$_H$) and the concomitant formation of NADPH. The latter was then quantified by directly measuring its 340 nm absorbance by applying the molar absorption coefficient ($\varepsilon$) NADPH$_{340}$ = 6220 M$^{-1}$ cm$^{-1}$.

The readings of the NADPH produced at each time point, for each substrate concentration, were fitted by linear regression and converted back into nmoles of GABA (or GABA-P$_H$) s$^{-1}$. The reaction rates at increasing substrate concentration were then fitted to the standard Michaelis–Menten equation as in GraphPad Prism 4.0. (GraphPad Software, San Diego, CA).

**Determination of minimum inhibitory concentration**. The MIC of *D,L*-Glu-γ-P$_H$, *L*-Glu-γ-P$_H$ and *D*-Glu-γ-P$_H$ that inhibits the growth of the *E. coli* K12 strain MG1655 was assessed using the broth dilution method[59] with some minor modifications, i.e., MIC$_{90}$ was calculated in this work, i.e., meaning that the indicated value is the concentration that inhibits 90% of the growth (as assessed at O.D.$_{600}$) at a specific time from the inoculum (22 h). MIC$_{90}$ is derived by the following equation: % inhibition = [1 − (OD$_{600}$treated/OD$_{600}$untreated)] × 100.

Briefly, overnight cultures (2 mL) of *E. coli* K12 strain MG1655 were centrifuged at 3500 rpm for 15 min at 15 °C and the cell pellets were resuspended in an equivalent volume of physiological solution (0.9% NaCl) and the O.D.$_{600}$ was brought to 1.0. The resuspended cells were inoculated (1:25) in 2 mL of Minimal

EG pH 7 and grown at 37 °C under moderate shaking up to O.D.$_{600}$ = 0.5 (7–8 h from the inoculum) and then diluted (1:25) in 3 mL of Minimal medium EG pH 7[56] and inoculated (1:10) in a 96-well microplate (Falcon™) previously set up with the appropriate serial dilutions of the compound(s) to be tested. The final volume in each well was 200 µL. The number of colony forming units/ml at time zero was between 0.5 and $1 \times 10^6$, following the CLSI guidelines for antimicrobial susceptibility testing[60]. For each compound tested, 2–4 independents assays were performed.

The microplate was incubated at 37 °C for 24 h and reading carried out on microplate readers (Sunrise, Tecan or Varioskan Lux, Thermo Scientific) and the OD$_{600}$ was automatically recorded every hour. Before each reading, the microplate was set to shake 10 s in order to resuspended the bacteria.

**Kinetic resolution of *D,L*-Glu-γ-P$_H$ by *Ec*GadB.** The reaction mixture (100 µL for analytical purposes or 2.5 mL for preparative purposes) consisted of 200 mM Pyridine/HCl buffer, pH 4.6, containing 1 mM PLP, 0.1 mM DTT, 0.4 M of *D,L*-Glu-γ-P$_H$ to which GadB was added to a final concentration of 1 µg µL$^{-1}$.

The analytical reactions (100 µL) were performed at 37 °C and at time intervals (0, 2, 5, 10, 20, 30, 60, 120 min). During the reaction, the pH was periodically adjusted to the value 4.6–5.0 by stepwise addition of 1 N HCl (total: 10 µL). At the indicated times, aliquots (5 µL) were transferred in 50 µL of 10 mM NaOH and then 2 µL analysed for GABA content with the GABase assay (see previous section). The experiment was performed in duplicate.

The preparative reaction was performed at 37 °C, adjusting the pH periodically to the value of 4.6–5.4 by adding 1 N HCl (total: 80 µL). At time intervals (0, 2, 5, 10, 20, 40, 60, 90, 120, 150, 180, 240, 300 min), aliquots (15 µL) were transferred in 10 µL of 250 mM NaOH, the precipitated protein was separated by centrifugation, D$_2$O (600 µL) was added to supernatant and $^{31}$P NMR spectra were measured on a Bruker 500 UltraShield™. The experiment was performed in triplicate.

**Molecular modelling.** All molecular modelling studies were performed on a MacPro dual 2.66 GHz Xeon running Ubuntu 14LTS. The *Ec*GadB structure was downloaded from the PDB data bank (http://www.rcsb.org/), PDB code 1PMM. We used a functional dimer of the hexameric enzyme, i.e., chains A and B. The aldimine PLP/Lys276 was bound at chain A, the PLP/*L*-Glu or *L*-Glu-γ-P$_H$ adducts were bound at chain B. The adducts structures were drawn by Maestro using the coordinates of the PLP and acetic acid (ACY) of the 1PMM structures. For the PLP-*L*-Glu external aldimine adduct the α-carboxy acid was deprotonated (carboxylate form) and the γ-carboxy acid was in a neutral from. For the PLP-*L*-Glu-γ-P$_H$ external aldimine adduct both the phosphinic and the α-carboxy acid groups were deprotonated. For both adducts, the pyridine and the aldimine nitrogen atoms were protonated. The MD was performed with the AMBER 12 suite[61]. For simulation, the protein was solvated in a periodic octahedron simulation box using TIP3P water molecules, providing a minimum of 10 Å of water between the protein surface and any periodic box edge. Ions were added to neutralise the charge of the total system. The solvent molecules and the ions were energy-minimised keeping the coordinates of the protein-ligand complex fixed (1000 cycle), and then the whole system was minimised keeping the adduct fixed (2000 cycle), then the all system was minimised without restraint (2000 cycle). Following minimisation, the entire system was heated to 298 K (2 ps). The production simulation was conducted at 298 K with constant pressure and periodic boundary condition (1 ns). Shake bond length condition was used (ntc = 2). Compounds were parametrised by Antechamber using BCC charges[62,63]. All the Asp, Glu and His residues of the binding site were titrated during the simulations keeping the protonation state fixed. The pH was kept at 4.5 by solvph = 4.5 and ntcnstph = 15,000 parameters in Amber production input file. Trajectories analysis was carried out by cpptraj programme[64]. The pictures reported in the manuscript were done with PyMOL (The PyMOL Molecular Graphics System, Version 2.0 Schrödinger, LLC).

**Reporting summary.** Further information on research design is available in the Nature Research Reporting Summary linked to this article.

## Data availability
The data that support the findings of this study are available from the corresponding author upon reasonable request.

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

## Acknowledgements

This work was supported by Sapienza University of Rome (Project RM11916B861B9985) to D.D.B., by Academy of Finland (Grant 315487) to J.V. and by the Russian Foundation of Basic Research (Grant No. 17–00–00395) to A.K. The authors thank Dr. A.O. Chizhov (N.D. Zelinsky Institute of Organic Chemistry, Russian Academy of Sciences, Moscow) for measurement of mass spectra.

## Author contributions

D.D.B. and A.K. conceived the study and interpreted the results. D.D.B., F.C., E.P., F.G. and A.K. performed experiments and analysed the data from the relevant experiments. J.V. and A.K. interpreted NMR spectra. A.C. performed the molecular modelling. D.D.B. and A.K. wrote the paper. E.P., F.G., A.C. and J.V. read the paper and provided feedback.

## Competing interests

The authors declare no competing interests.
