## [Peer Review File · Communications Chemistry]

Reviewers' comments:

Reviewer #1 (Remarks to the Author):

The reviewed paper considers synthesis of phosphinic acid mimetic of glutamic acid and study of its influence on glutamate decarboxylase from *Escherichia coli*. Quite obvious finding that only L-isomer of this mimetic is hydrolysed stimulated the studies on the use of this decarboxylase for preparation of D-isomer of the studied compounds. It is worth to note that decarboxylation of this mimetic provides known effector of neurotransmission - phosphinic analogue of GABA.

Thus, the paper is of interest to chemists (both medicinal and organic) and biochemists dealing with organophosphorus compounds.

Paper is well composed and well written. Examination of Supplementary material is convincing. In my opinion it can be published as it is.

Reviewer #2 (Remarks to the Author):

see separate file

The paper by De Biase et al. describes the synthesis of a phosphinic acid derivative of glutamic acid, and its decarboxylation by the glutamate α -decarboxylase from *E. coli*. The authors convincingly show that this derivative is converted by the enzyme in an enantioselective fashion, while its oxidised phosphonic acid analogue is not accepted by the enzyme. A rationale for this is given by molecular docking. Kinetic parameters for the reaction have been determined. The reaction has been performed on preparative scale (though still very small) enabling the authors to unambiguously determine the structure of the products. The products of this reaction and subsequent enzymatic conversions are of pharmaceutical interest.

The work presented is novel and in my view it is interesting enough to be publishable in this journal, after some minor changes have been implemented, see below.

Scientific comments:

- The stereochemical drawings have to be changed. Groups pointing upwards should be annotated with filled wedges, groups pointing downwards with dashes. Specifically, the NH_2 in *D*-Glu- γ - P_H in Figure 2 should have dashes, and also all NH_2 's in Figure 1.
- Line 140 and Fig. 1 in SI: please provide literature evidence for the flattened tetrahedral structure of the phosphinic group, especially because modelling in Fig. 3 shows it as a normal tetrahedron.
- Line 98: the authors claim that *L*-Glu- γ - P_H has antibacterial properties, but that does not say anything; sodium chloride is also antibacterial. It all depends on the MIC_{90} compared to well-known commercial antibacterials. Only if the MIC_{90} is in the same range a compound should be called an antibacterial. The results in Table 2 should be put in perspective in this way.
- The results in Table 2 are puzzling. Why is there such a large range (uncertainty?) in the MIC_{90} of the racemate? If the *L*-isomer has 50-33% of the MIC_{90} value of the racemate, then I expect the *D*-value to have twice-thrice the MIC_{90} value of the racemate, rather than an undetectably high MIC_{90} . Or is this line of thinking too simple (I am not a microbiologist)?
- The discussion is a bit lengthy, please shorten

Textual comments:

- Line 208 and elsewhere: GABase, not Gabase
- 279: with respect to the amount
- 321: making it impossible
- 347: a hardly detectable activity...
- 411: having *C-P-H* bonds....
- 645: names of authors is weird

Authors point-by-point reply to

Reviewers' comments:

Reviewer #1 (Remarks to the Author):

The reviewed paper considers synthesis of phosphinic acid mimetic of glutamic acid and study of its influence on glutamate decarboxylase from *Escherichia coli*. Quite obvious finding that only L-isomer of this mimetic is hydrolysed stimulated the studies on the use of this decarboxylase for preparation of D-isomer of the studied compounds. It is worth to note that decarboxylation of this mimetic provides known effector of neurotransmission - phosphinic analogue of GABA.

Thus, the paper is of interest to chemists (both medicinal and organic) and biochemists dealing with organophosphorus compounds.

Paper is well composed and well written. Examination of Supplementary material is convincing. In my opinion it can be published as it is.

We thank the reviewer for the positive evaluation of our work and the appreciation that it will be of interest and, we hope, of inspiration to chemists and biochemists working in this field.

Reviewer #2 (Remarks to the Author):

The paper by De Biase et al. describes the synthesis of a phosphinic acid derivative of glutamic acid, and its decarboxylation by the glutamate α -decarboxylase from *E. coli*. The authors convincingly show that this derivative is converted by the enzyme in an enantioselective fashion, while its oxidised phosphonic acid analogue is not accepted by the enzyme. A rationale for this is given by molecular docking. Kinetic parameters for the reaction have been determined. The reaction has been performed on preparative scale (though still very small) enabling the authors to unambiguously determine the structure of the products. The products of this reaction and subsequent enzymatic conversions are of pharmaceutical interest.

The work presented is novel and in my view it is interesting enough to be publishable in this journal, after some minor changes have been implemented, see below.

We thank the reviewer for the positive evaluation of the novelty of our work and the appreciation that it will be of interest for Communications Chemistry readership. Below our reply to all the points raised by the reviewer.

Scientific comments:

- The stereochemical drawings have to be changed. Groups pointing upwards should be annotated with filled wedges, groups pointing downwards with dashes. Specifically, the NH₂ in D-Glu- γ -PH in Figure 2 should have dashes, and also all NH₂'s in Figure 1.

Following the suggestions of the reviewer, we changed the style of NH₂-bonds in Fig.2 into dashed one. Besides, to underline that naturally occurring DMPT and PT have L-configuration we used filled NH₂-bonds in Fig.1. In this way this Figure is more aligned with the ms text.

Finally, for consistency, we modified accordingly also the drawing in Supplementary Fig. 7.

- Line 140 and Fig. 1 in SI: please provide literature evidence for the flattened tetrahedral structure of the phosphinic group, especially because modelling in Fig. 3 shows it as a normal tetrahedron.

*We have added a new reference (Ref. 35 in the revised ms and Ref. 1 in Supplementary Information): Schwalbe C.H., Freeman S., DasGupta M. "2-Amino-2-carboxyethylphosphinic acid monohydrate". *Acta Cryst.*, C49, 1826-1828 (1993). Though we had a couple more (<https://doi.org/10.1039/DT9920000939>; <http://dx.doi.org/10.1080/10426509408021464>), we think that Schwalbe et al. is the most appropriate to cite in our ms because it refers to the β -phosphinic analogue of aspartate, a molecule closely related to L-Glu- γ -P_H.*

- Line 98: the authors claim that L-Glu- γ -PH has antibacterial properties, but that does not say anything; sodium chloride is also antibacterial. It all depends on the MIC₉₀ compared to well-known commercial antibacterials. Only if the MIC₉₀ is in the same range a compound should be called an antibacterial. The results in Table 2 should be put in perspective in this way. *We thank the reviewer for raising this point. We had this information available for Ampicillin, but for some reason forgot to include it in the first submitted version. Now Ampicillin is included in Table 2 and we refer to it in the revised text. As it can be appreciated, L-Glu- γ -P_H is in the same MIC range (i.e. <10 μ g/ml) of Ampicillin. As a personal communication we can also add here that, under our assays conditions, L-Glu- γ -P_H is effective (and MIC unaffected) also in a strain carrying resistance to Ampicillin and Kanamycin. These and other results (a work currently underway) point to a mechanism of action different from that of other antibiotics available in the market and this adds a new level of interest for investigating the cellular targets and the mechanism of action of this molecule.*
- The results in Table 2 are puzzling. Why is there such a large range (uncertainty?) in the MIC₉₀ of the racemate? If the L-isomer has 50-33% of the MIC₉₀ value of the racemate, then I expect the D-value to have twice-thrice the MIC₉₀ value of the racemate, rather than an undetectably high MIC₉₀. Or is this line of thinking too simple (I am not a microbiologist)? *It is indeed important for us to be sure that this point is clear even to a non-microbiologist: the text following the table has been changed to clarify the point and also a reference added (Ref. 48 in the revised ms). First of all we must say that for microbiologists the provided range is not large because MIC experiments are typically performed by using serial dilutions (1:2 or sometimes 1:10). We agree that MIC values are typically reported as precise values, but for Editor and Reviewer perusal we provide here 2 literature references and some relevant tables where MIC values are provided as a range (even very large sometimes).*

Regarding our work, why the MIC₉₀ of the racemate is not exactly twice the value of the L-Glu γ -P_H? it is not easy to provide an exact explanation at present. We are sure that the D-enantiomer is not active as such (as confirmed by testing its activity as a pure substance), but we cannot exclude that the D-enantiomer when present in D,L-Glu γ -P_H affects to some extent the

entrance of the L-enantiomer in the cell and this may be the cause of a the slight increase of the MIC. Now we provide Ref. 48 in support to this possibility.

Further studies will hopefully allow us to clarify this point

- The discussion is a bit lengthy, please shorten.

We re-read the discussion and a sentence (3 lines) was removed. Overall we feel that we discuss the results in a perspective way, with only very limited overlap with the results section and did not find the Discussion lengthy (2 pages double spacing + Fig. 6). In the Discussion we introduced 5 more references (49-53) and we believe all are relevant to our work and need to be maintained. We hope that the reviewer is happy with the applied changes.

Textual comments:

- Line 208 and elsewhere: GABase,not Gabase. *Absolutely right. Thanks. Amended*
- 279: with respect to the amount. *done*
- 321: making it impossible *done*
- 347: a hardly detectable activity... *done*
- 411: having C-P-H bonds.... *done*
- 645: names of authors is weird. *Apologies for the oversight. Amended*